# Development of Water Level Prediction Improvement Method Using Multivariate Time Series Data by GRU Model



Kidoo Park [1], Yeongjeong Seong [2,*], Younghun Jung [2], Ilro Youn [3] and Cheon Kyu Choi [4]

1 Emergency Management Institute, Faculty of Engineering, Kyungpook National University, Sangju 37224, Gyeongbuk, Republic of Korea
2 Faculty of Engineering, Department of Advanced Science and Technology Convergence, Kyungpook National University, Sangju 37224, Gyeongbuk, Republic of Korea
3 Department of Construction and Disaster Prevention Engineering, Faculty of Engineering, Kyungpook National University, Sangju 37224, Gyeongbuk, Republic of Korea
4 Department of Hydro Science and Engineering Research, Korea Institute of Civil Engineering and Building Technology, Goyang 10223, Gyeonggi, Republic of Korea
* Correspondence: bnmjkl31@knu.ac.kr; Tel.: +82-54-530-1256

**Abstract:** The methods for improving the accuracy of water level prediction were proposed in this study by selecting the Gated Recurrent Unit (GRU) model, which is effective for multivariate learning at the Paldang Bridge station in Han River, South Korea, where the water level fluctuates seasonally. The hydrological data (i.e., water level and flow rate) for Paldang Bridge station were entered into the GRU model; the data were provided by the Water Resources Management Information System (WAMIS), and the meteorological data for Seoul Meteorological Observatory and Yangpyeong Meteorological Observatory were provided through the Korea Meteorological Administration. Correlation analysis was used to select the training data for hydrological and meteorological data. Important input data affecting the daily water level (DWL) were daily flow rate (DFR), daily vapor pressure (DVP), daily dew point temperature (DDPT), and 1 h max precipitation (1HP), and were used as the multivariate learning data for water level prediction. However, the DWL prediction accuracy did not improve even if the meteorological data from a single meteorological observatory far from the DWL prediction point were used as the multivariate learning data. Therefore, in this study, methods for improving the predictive accuracy of DWL through multivariate learning that effectively utilize meteorological data from each meteorological observatory were presented. First, it was a method of arithmetically averaging meteorological data for two meteorological observatories and using it as the multivariate learning data for the GRU model. Second, a method was proposed to use the meteorological data of the two meteorological observatories as multivariate learning data by weighted averaging the distances from each meteorological observatory to the water level prediction point. Therefore, in this study, improved water level prediction results were obtained even if data with some correlation between meteorological data provided by two meteorological observatories located far from the water level prediction point were used.

**Keywords:** water level; GRU; meteorological data; multivariate learning; correlation; meteorological observatory

## 1. Introduction

In the field of water resource engineering, data models using artificial intelligence have developed remarkably over the past 20 years. Traditionally, as a data-driven model for water level prediction in the field of water resource engineering, the Auto-Regressive Integrated Moving Average (ARIMA) model has been used to predict the monthly average water level, which is a linear time series data [1]. However, in the case of the ARIMA model prediction results, the prediction accuracy of hydrological time series data with nonlinear characteristics was significantly reduced compared to that of linear time series data [2]. In

addition, in the case of the ARIMA model, it was used to predict the groundwater level, but there was a limitation in that the hydrological correlation between the rainfall and the groundwater level could not be properly considered [3].

In addition, in the case of the fuzzy-Neural Network (NN) system, one of the representative data-driven models, it was predictable with high accuracy in predicting the reservoir water levels [4,5]. However, even in the case of the fuzzy-NN system, the predictive accuracy of its model was greatly reduced in the prediction of the nonlinear time series groundwater level [3].

Studies have been steadily conducted to improve accuracy in terms of water level prediction, where rapid changes in water quantity such as flooding occur in the field of water resource engineering. Since 2015, with the start of the Fourth Industrial Revolution, the predictive accuracy of the Deep Neural Network (DNN) model has improved dramatically in the field of image analysis and natural language processing. With the recent active development of the Deep Learning (DL) model, it has been steadily studied as a model that can replace the previous data-driven models (i.e., ARIMA, fuzzy-NN) for predicting the existing hydrological time series. Therefore, the studies in the hydrological and water resources fields have also begun to be actively used in predicting hydrological data using DNN [6,7]. In the studies of the DNN model of the last decade, high accuracy prediction was possible in the groundwater level prediction [8], river water level prediction [9–11], and reservoir water level prediction [12–14] using time series data. The prediction of the hydrological data using DL models could obtain high prediction accuracy only when there was little change in water level or flow rate or when its change was very small. However, in a situation where a rapid flood occurred and a rapid rise in the water level was predicted, accurate prediction results for the high water level could not be obtained [15]. Therefore, the case of rapidly changing water level prediction occurring during the flood season has been recognized as a limitation of the DNN models.

Nevertheless, in order to properly predict rapid hydrological changes, it has been continuously developed into a groundwater level prediction accuracy analysis [16] and peak water level prediction due to tidal changes [17,18] using various DNN models. Among the various DNN models, CNN, simple RNN, LSTM, and GRU models were used to evaluate the prediction accuracy of the model through flow prediction varying from low to high flow rates [15]. The various studies have been conducted on the improvement of model accuracy through research on the selection of RNN models, the composition of models, the sequence length of time series data, and the ratio and length of input and prediction data [19,20]. As a result of comparing the hydrological prediction performance, the GRU model and the LSTM model were evaluated as appropriate for the hydrological time series prediction model, which can ensure the appropriate accuracy of the highly variable flow rate and water level [15,21]. In addition, in order to predict the flow rate of rivers with large flow fluctuations using the LSTM and GRU, a study was also conducted to indirectly increase the prediction accuracy of the flow rate by using the water level flow rate rating curve [22].

The prediction results of the LSTM and GRU were improved when multivariate learning with greater correlation was performed compared to when only the univariate hydrologic data were trained for the prediction of rapidly changing hydrologic data [21]. Even in hydrological time series prediction using multivariate learning data, the prediction results of the GRU were more accurate than those of the LSTM [21]. When performing water level prediction through the GRU model using a highly fluctuating water level, the accuracy of the predicted time series water level is very low if only the water level is trained as a single input variable of the model. Therefore, the accuracy of water level prediction could be improved by using meteorological data observed in the meteorological observatory near the water level prediction point [21]. However, it is not easy to use such data from meteorological stations that can directly affect the river where the water level prediction point is located. In other words, if only the meteorological data of a single

meteorological observatory located at a long distance from the water level prediction point are used, the accuracy of the predicted water level cannot be improved.

The aim of this study was to propose a method of improving the performance of a GRU model to predict water level using multivariate meteorological data from two distant observatories. First, two meteorological observatories far from the water level prediction point were selected, and meteorological data with the correlation between the meteorological data as well as the water level were selected through correlation analysis of the data from the two observatories. Second, it was used as multivariate input data for the GRU model using meteorological data from two meteorological observatories with small, selected correlation. Because the distance between the two meteorological observatories is far from each other, the characteristics of the meteorological data are different, but if the arithmetic average or distance-weighted average of the location of the water level station was used, meteorological data with improved correlation could be obtained. The first method used in this study is a method of arithmetically averaging meteorological data of two meteorological observatories and inputting them as multivariate learning data. The second method involves inputting multivariate meteorological data obtained by the weighted average of the distance from the water level prediction point to the meteorological observatory as learning data of the GRU model. Therefore, even if the distance between the observation point and the meteorological observatories is significantly apart, the data from the meteorological stations can be effectively utilized to produce the results of improved water level prediction accuracy at the water level prediction point.

## 2. Methods

### 2.1. Gated Recurrent Unit (GRU)

As shown in Figure 1, GRU plays a similar role as long short-term memory (LSTM) but it is computationally efficient because it consists of a simpler structure. The input gate and forget gate are combined and simplified into an update gate [15,21,22]. GRU has two activation functions and one tanh function. Therefore, GRU is capable of long-term memory like LSTM, but it has the advantage of having fewer parameters and faster training speed than LSTM.

$$r_t = \sigma(W_r \cdot [h_{t-1}, \ x_t] + b_r) \tag{1}$$

$$z_t = \sigma(W_z \cdot [h_{t-1}, \ x_t] + b_z) \tag{2}$$

$$\overline{h}_t = \tan h(W \cdot [h_{t-1}, \ x_t] + b) \tag{3}$$

where $r$ and $z$ are the reset and update gates, respectively. Reset gate aims to reset past data and outputs a value between 0 and 1, which is the value of how much past data will be reset through the activation function. The update gate determines the rate of past and present information updates and the output value $z_t$ determines the amount of data to be exported at this point in time. $1 - z_t$ is the amount of data to be forgotten.

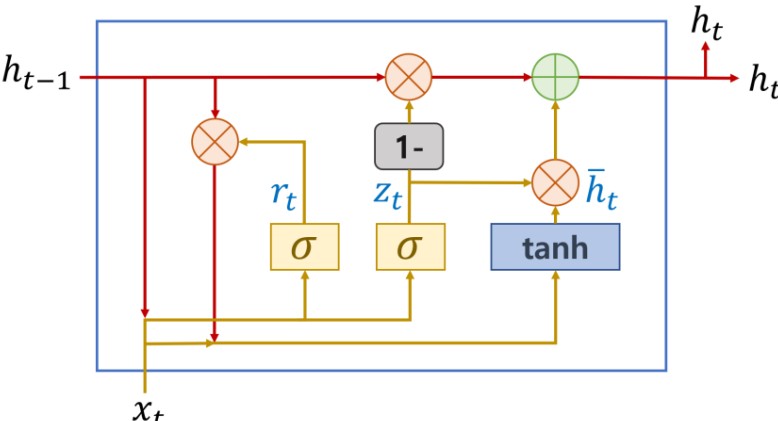

**Figure 1.** Gated Recurrent Unit (GRU).

## 2.2. Model Performance Indicators

Equations (5)–(8) were used as evaluation criteria for the GRU model to evaluate the accuracy and performance of the model. The closer the Mean Square Error (MSE), and Mean Square Error (RMSE) are to 0, the better the performance of the model. As the Nash–Sutcliffe model efficiency coefficient (NSE). The determination coefficient ($R^2$) are close to 1, the performance of the model is improved.

(1)　Mean Squared Error (MSE)

MSE measures the mean of the squared differences between the predictions and the actual observations [15,21–23].

$$MSE = \frac{1}{N} \sum_{i=1}^{N} (x_i - y_i)^2 \tag{4}$$

where $x_i$ are the observed values of the variables, $y_i$ are the predicted values and $N$ is the number of data.

(2)　Root Mean Squared Error (RMSE)

RMSE is the square root of the average of squared differences between predictions and actual observations [15,21–25].

$$RMSE = \sqrt{\frac{\sum_{i=1}^{N} (x_i - y_i)^2}{N}} \tag{5}$$

(3)　Coefficient of determination ($R^2$)

The coefficient of determination $R^2$ is a measure of the goodness of fit of the statistical model [15,21–23,26,27].

$$R^2 = 1 - \frac{\sum_{i=1}^{N} (x_i - \hat{y}_i)^2}{\sum_{i=1}^{N} (x_i - \overline{x})^2} \tag{6}$$

where $\hat{y}_i$ are the predicted values from a statistical model and $\overline{x}$ is the mean of observed values of the variables.

(4)　Nash–Sutcliffe model efficiency coefficient (NSE)

NSE quantifies how well model simulations can predict outcome variables [15,21–24,26,27].

$$NSE = 1 - \frac{\sum_{i=1}^{N} (x_i - y_i)^2}{\sum_{i=1}^{N} (x_i - \overline{x})^2} \tag{7}$$

As shown in Table 1, it is inappropriate to adopt model results if $R^2$ and $NSE$ are less than 0.5, model adoption is possible if $R^2$ and $NSE$ are greater than 0.5 and less than 0.65, model adoption is good if $R^2$ and $NSE$ are greater than 0.65 and less than 0.75, and if $R^2$ and $NSE$ are over 0.75, it is very good to adopt a model [15,21–24,26,28].

**Table 1.** Performance ratings for adopted statistics.

| Performance Rating | $R^2$ | $NSE$ |
|---|---|---|
| Very good | $0.75 < R^2 \leq 1.00$ | $0.75 < NSE \leq 1.00$ |
| Good | $0.65 < R^2 \leq 0.75$ | $0.65 < NSE \leq 0.75$ |
| Satisfactory | $0.50 < R^2 \leq 0.65$ | $0.50 < NSE \leq 0.65$ |
| Unsatisfactory | $R^2 \leq 0.50$ | $NSE \leq 0.50$ |

## 2.3. Application of Models

As shown in Figure 2, this study attempts to compare the accuracy of rapidly varying water level prediction and the performance of models using the GRU model. The data

input and output to time series models basically uses water levels. However, since water level data alone are limited in accurately predicting rapidly changing water levels, this study attempts to analyze the accuracy of the predicted water level by using hydrological data (i.e., flow rate, dew-point temperature, vapor pressure, and precipitation) collected at nearby observatories, which are correlated with the water level data. When using only a single hydrological time series input data, the correlation between hydrological input data and water levels collected from nearby meteorological observatories was analyzed to overcome the limitation of not accurately providing rapidly changing water levels. Therefore, the purpose of this study was to accurately predict water level by considering the multivariate hydrological time series data according to the spatial distribution of the two meteorological observatories.

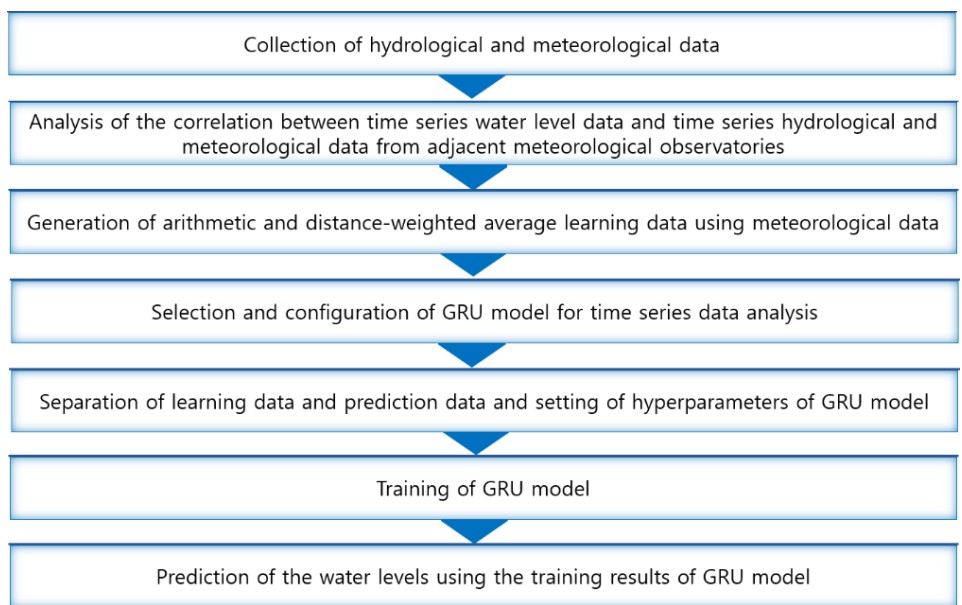

**Figure 2.** Flowchart of water level prediction using GRU model.

### 3. Study Area and Data

*3.1. Study Area*

The Han River in the urbanized Han River basin was selected (Figure 3). The Han River basin is located in the central part of the Korean Peninsula and spans 36°30′ to 38°55′ N latitude, 126°24′ to 129°02′ E longitude. The Han River is the largest river in South Korea with the basin area of 25,953.60 km², the total length of 494.44 km, the average width of 72.35 km, and the shape coefficient of 0.146. The Han River basin is a multi-form basin mixed with dendritic and facsimile forms [15,21,29].

As shown in Figure 3, after the South Han River and North Han River join the Paldang Reservoir, the main stream of the Han River is formed from the Paldang Dam. The Paldang Bridge Observatory is located directly downstream of Paldang Dam, where the main stream of the Han River begins. The river topographic characteristics of the section between Jamsil Weir and Paldang Dam of the Han River are as follows. The width of the river is 303–1693 m, the width of the channel is 277–1332 m, and the average slope of the river is 1/10100. The study site is a section in the middle of the Han River that flows through major cities of Seoul and Gyeonggi-do. The study area has experienced rapid urbanization. A waterfront space and an ecological wetland are found in the area.

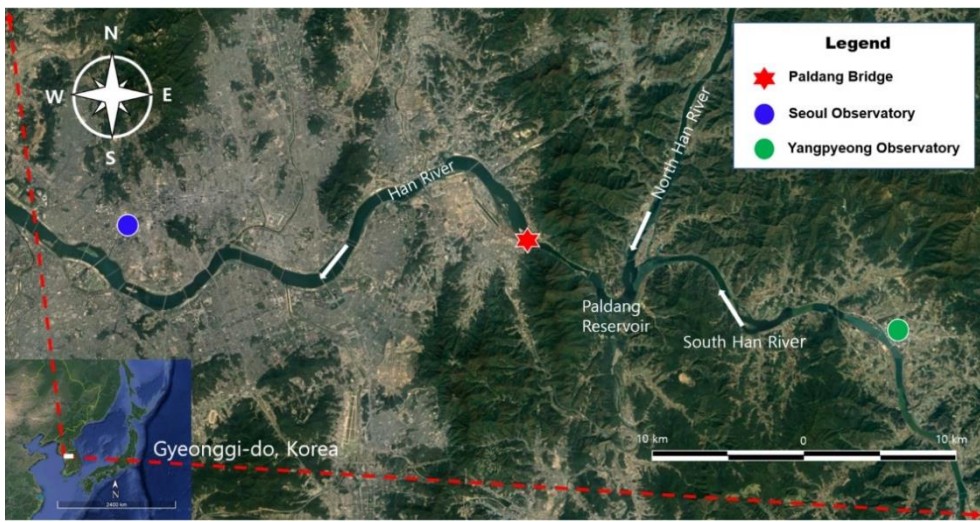

**Figure 3.** Map of the Han River basin in Korea (Google Earth [30]).

*3.2. Hydrologic and Meteorological Data*

In this study, hydrological data (i.e., water level, flow rate) for Paldang Bridge Station located upstream of the Han River observed by the Ministry of Environment of Korea were used (Figure 3). The weather data of Seoul Observatory and Yangpyeong Observatory were used at the two nearest Korea Meteorological Administration (KMA) observatories from Paldang Bridge Station (Figure 3) [31]. As shown in Figure 3, the distance between the two meteorological observatories is as follows: (1) It is located at the downstream of the Han River 24.1 km from the Seoul Observatory to the Paldang Bridge station. (2) In the case of Yangpyeong Observatory, it is located at the upstream of the South Han River 23.5 km to the Paldang Bridge station. (3) The water level data at the flow measuring station were obtained using data from the Water Resources Management Information System (WAMIS) website of the Ministry of Environment [32].

(1)　Daily water level

In this study, the longer the training data, the better the overall prediction results of the GRU model. So, the total length of the data used for learning and prediction was 2 years and 7 months and relatively short data were used to verify GRU model.

As shown in Table 2 and Figure 4, the average water level observed at Paldang Bridge Station was based on real-time observation data from 1 January 2018 to 31 July 2020. In order to analyze the characteristics of water levels, the average, minimum, maximum water levels were statistically analyzed as shown in Table 2.

**Table 2.** Statistical characteristics of water levels at the Paldang Bridge station (period: 2018–2020).

| Minimum Water Level (EL.m) | Maximum Water Level (EL.m) | Average Water Level (EL.m) | Standard Deviation of Water Level (EL.m) |
|---|---|---|---|
| 0.960 | 4.230 | 1.190 | 0.328 |

As shown in Figure 4, the selected daily average time series water level data was divided into learning data and model prediction data. Using the results of previous studies [15], which obtained the best accuracy, the optimal length of training data and prediction data was divided into 74.9% and 25.1%, respectively, from the length of the all-time series data. As shown in Figure 4, the data with black dotted lines were used as training data for GRU model, and solid red line data were used as predictive data for its model.

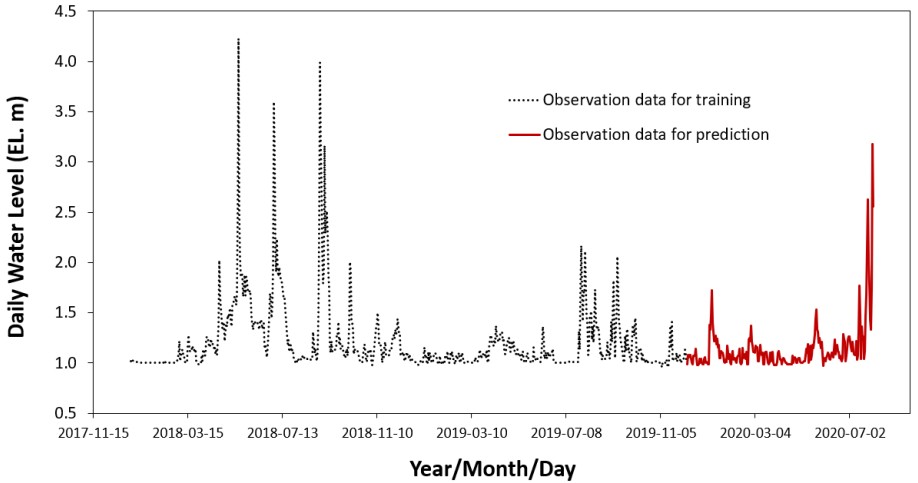

**Figure 4.** Time series daily water level at the Paldang Bridge station (period: 2018–2020).

(2) Hydrological and meteorological data

As shown in Table 3, the correlation between meteorological and hydrological data measured at two nearby observation stations (i.e., Seoul Observatory and Yangpyeong Observatory) was analyzed to select input data suitable for DWL prediction at Paldang Bridge.

**Table 3.** Correlation between hydrological and meteorological data.

| Variable | DWL (EL.m) | DFR (m³/s) | S_1HP (mm) | S_DDPT (°C) | S_DVP (hPa) | Y_1HP (mm) | Y_DDPT (°C) | Y_DVP (hPa) |
|---|---|---|---|---|---|---|---|---|
| DWL (EL.m) | 1.0000 | 0.9735 | 0.2350 | 0.3254 | 0.3328 | 0.3415 | 0.3607 | 0.3809 |
| DFR (m³/s) | 0.9735 | 1.0000 | 0.2512 | 0.2687 | 0.2790 | 0.3281 | 0.3034 | 0.3291 |
| S_1HP (mm) | 0.2350 | 0.2512 | 1.0000 | 0.3047 | 0.3770 | 0.1555 | 0.2402 | 0.2652 |
| S_DDPT (°C) | 0.3254 | 0.2687 | 0.3047 | 1.0000 | 0.9420 | 0.2310 | 0.8407 | 0.8387 |
| S_DVP (hPa) | 0.3328 | 0.2790 | 0.3770 | 0.9420 | 1.0000 | 0.2550 | 0.8228 | 0.8795 |
| Y_1HP (mm) | 0.3415 | 0.3281 | 0.1555 | 0.2310 | 0.2550 | 1.0000 | 0.2796 | 0.3419 |
| Y_DDPT (°C) | 0.3607 | 0.3034 | 0.2402 | 0.8407 | 0.8228 | 0.2796 | 1.0000 | 0.9472 |
| Y_DVP (hPa) | 0.3809 | 0.3291 | 0.2652 | 0.8387 | 0.8795 | 0.3419 | 0.9472 | 1.0000 |

DWL, daily water level; DFR, daily flow rate; S_DVP, daily vapor pressure at Seoul Observatory; S_DDPT, daily dew-point temperature at Seoul Observatory; S_1HP: 1 h max precipitation at Seoul Observatory; Y_DVP, daily vapor pressure at Yangpyeong Observatory; Y_DDPT, daily dew-point temperature at Yangpyeong Observatory; Y_1HP: 1 h max precipitation at Yangpyeong Observatory.

First, the correlations between the daily water level (DWL) and the collected hydrological variables, such as daily flow rate (DFR), S_1HP (1-h max precipitation), S_DDPT (daily dew-point temperature), and S_DVP (daily vapor pressure) at Seoul Observatory, and Y_1HP, Y_DDPT, and Y_DVP at Yangpyeong Observatory, were analyzed. The hydrological variable with the largest correlation with the DWL to be predicted was daily flow rate (DFR), which was 0.9735, and the correlation between DWL and DFR was very high. The correlation coefficients for the hydrological data (i.e., S_DDPT, S_DVP, Y_1HP, Y_DDPT, and Y_DVP) of Seoul and Yangpyeong Observatories are 0.3254–0.3809, which are correlated with the water level. However, the S_1HP was 0.2350, which had a very low correlation with the DWL. The reason is that Yangpyeong Observatory is located right next to the South Han River, but Seoul Observatory is located in downtown Seoul under the Namsan Mountain, causing heterogeneity of weather phenomena due to topographical factors.

Second, the correlation between the meteorological data of the two meteorological observatories was 0.1555, which was very low because the precipitation pattern was

significantly different due to the difference in topographical characteristics. The correlation coefficients among S_DDPT, S_DVP, Y_DDPT, and Y_DVP were 0.8228–0.8795, which were very high. However, each of S_1HP and Y_1HP had a low correlation of 0.2310–0.2652 with other meteorological data.

*3.3. Model Composition*

(1)   Composition of GRU model

In this study, Python version 3.9.12 [33], an open source programming language and TensorFlow version 2.10.0 [34], a deep learning library, were used. The detailed model compositions such as the shape of neurons and the composition of layers used as GRU models are presented in Table 4. The GRU model consisted of one input layer, two hidden layers, one dropout layer, and two dense layers [15,21,22].

**Table 4.** Configuration and hyperparameters of GRU model.

| Activation Function | Input Layer | Hidden Layer 1 | Dropout | Hidden Layer 2 | Dense Layer 1 | Dense Layer 2 |
|---|---|---|---|---|---|---|
| ReLU | GRU | GRU 50 units | 0.25 | GRU 50 units | 25 units | 1 unit |

In the GRU model, sequence length (SL) was trained for 14 days to improve the prediction accuracy, 74.9% of all hydrological and meteorological data were used as the training data, and 25.1% of the remaining data not used for training were used as the prediction data [15,21,22]. A total of 600 epochs were trained for sufficient learning of the data, the optimizer used was the Adam optimizer, and the cost function was Mean Square Error (MSE).

(2)   Composition of training data in GRU model

Table 5 shows the input and output data of the GRU model. Using the GRU model, the hydrological and meteorological data for one or two meteorological observatories were combined to predict the high accuracy DWL at the Paldang Bridge point. In order to compare the accuracy of DWL prediction on Paldang Bridge according to the location characteristics of two meteorological observatories, multivariate input data (i.e., DFR, S_DVP, S_DDPT, S_1HP, Y_DVP, Y_DDPT, and Y_1HP) were constructed as shown in Table 5 by combining meteorological data with different precipitation patterns due to topography. The input variables used in the learning data were composed of 1 to 8 variables, and DWL was predicted and accuracy was evaluated through the GRU model.

**Table 5.** Composition of input and output data of GRU model.

| Number of Input Variables | Training Data | Prediction Data |
|---|---|---|
| 1 | DWL | DWL |
| 5 | DWL, DFR, S_DVP, S_DDPT, S_1HP | |
| 5 | DWL, DFR, Y_DVP, Y_DDPT, Y_1HP | |
| 8 | DWL, DFR, * equal(S_DVP, S_DDPT, S_1HP, Y_DVP, Y_DDPT, Y_1HP) | |
| 8 | DWL, DFR, ** distance(S_DVP, S_DDPT, S_1HP, Y_DVP, Y_DDPT, Y_1HP) | |

* considering the data of Seoul Observatory and Yangpyeong Observatory equally; ** considering the weighted average of the data of the Seoul and Yangpyeong Observatories on the distance

## 4. Results

### 4.1. Results on Training and Prediction Using Water Levels as Univariate Input Data

The learning and prediction results for the GRU model using univariate time series DWL training data are shown in Figure 5 and Table 6. The red circles in Figure 5a,c represent the training results, and the green circles in Figure 5a,c represent the prediction results. Figure 5b,d show the prediction accuracy of the model as $R^2$.

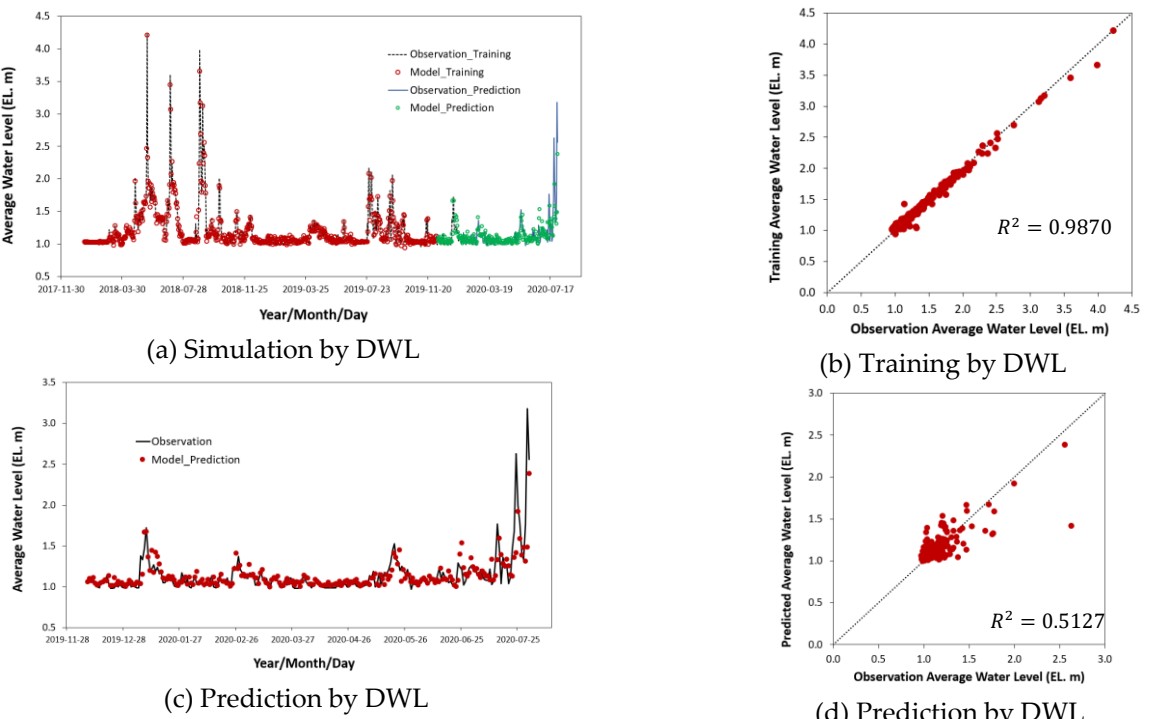

**Figure 5.** Observed and computed total time series water levels (**a**,**c**) as univariate input data, and $R^2$ for training (**b**) and $R^2$ for prediction (**d**) of GRU model.

**Table 6.** Comparison of univariate model performance.

| Variable | Computational State | MSE (EL.m) | RMSE (EL.m) | $R^2$ | NSE |
|----------|---------------------|------------|-------------|-------|-----|
| DWL | Training | $7.153 \times 10^{-6}$ | 0.0027 | 0.9889 | 0.9870 |
| | Prediction | $1.629 \times 10^{-5}$ | 0.0040 | 0.5179 | 0.5129 |

As shown in Table 6, when univariate DWL was trained as input data for the GRU model, the learning results of the model for DWL were 0.9889 for $R^2$ and 0.9870 for NSE, respectively, which was evaluated as "very good". However, the prediction results of the DWL data were 0.5179 for $R^2$ and 0.5129 for NSE, respectively, and the accuracy of the water level prediction was slightly "satisfactory". As shown in Figure 5b,d, the predicted accuracy of DWL was less than the observed value of DWL during peak water flow.

### 4.2. Results on Training and Prediction Using Water Levels and Multivariate Input Data

Figure 6 and Table 7 presented the results of predicting DWL by combining the flow rate and multivariate data from one or two meteorological observatories for DWL prediction. As shown in Table 7, the five and eight multivariate learning results of the GRU model were $NSE = 0.9807 - 0.9923$, indicating optimal ("very good") training results. However, even if multivariate input data were used considering only one meteorological observatory located at a considerable distance of 23.5 km or 24.1 km from the Paldang

Bridge station, the prediction results of DWL did not improve as $NSE = 0.5941 - 0.5976$, corresponds to "satisfactory".

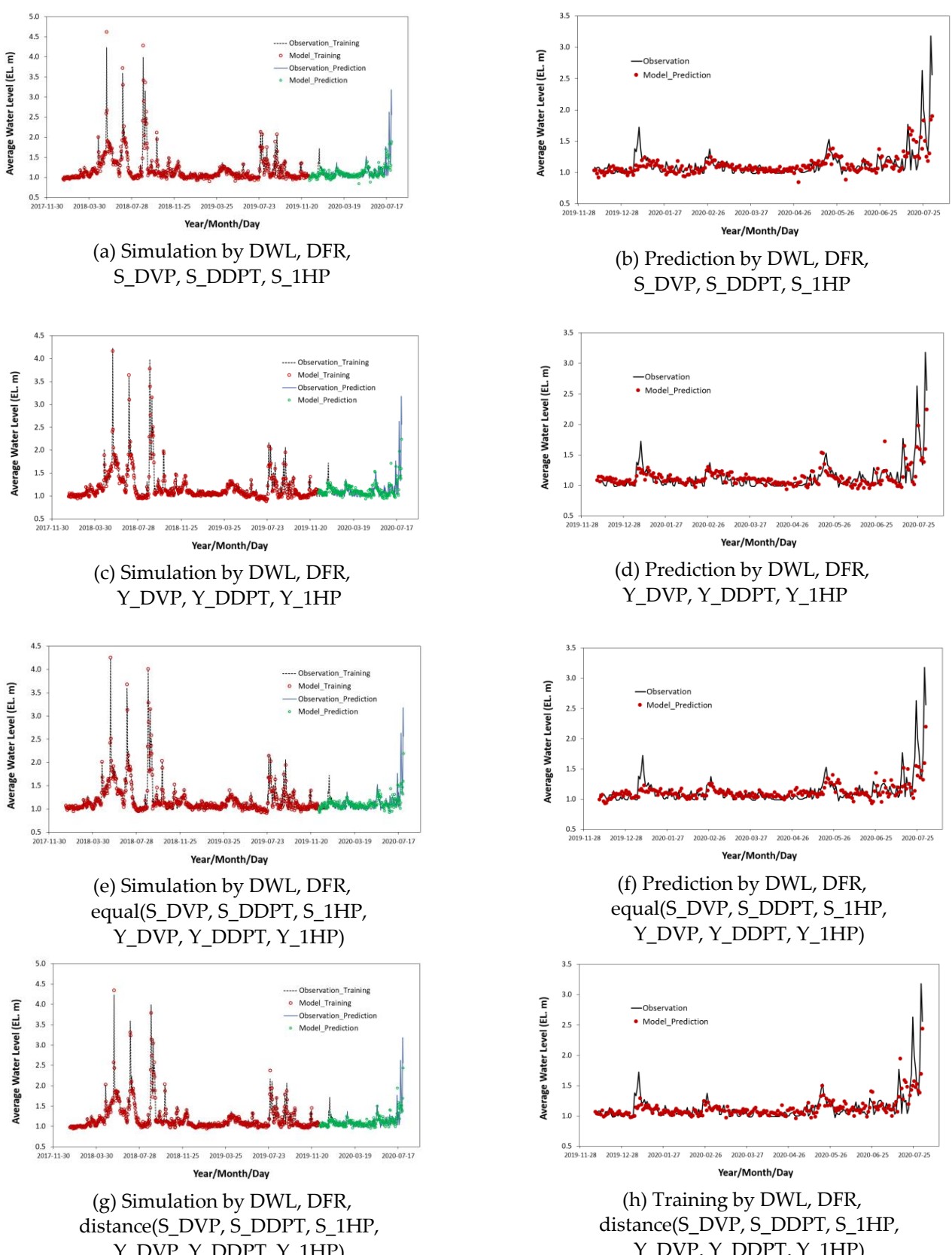

(a) Simulation by DWL, DFR, S_DVP, S_DDPT, S_1HP

(b) Prediction by DWL, DFR, S_DVP, S_DDPT, S_1HP

(c) Simulation by DWL, DFR, Y_DVP, Y_DDPT, Y_1HP

(d) Prediction by DWL, DFR, Y_DVP, Y_DDPT, Y_1HP

(e) Simulation by DWL, DFR, equal(S_DVP, S_DDPT, S_1HP, Y_DVP, Y_DDPT, Y_1HP)

(f) Prediction by DWL, DFR, equal(S_DVP, S_DDPT, S_1HP, Y_DVP, Y_DDPT, Y_1HP)

(g) Simulation by DWL, DFR, distance(S_DVP, S_DDPT, S_1HP, Y_DVP, Y_DDPT, Y_1HP)

(h) Training by DWL, DFR, distance(S_DVP, S_DDPT, S_1HP, Y_DVP, Y_DDPT, Y_1HP)

**Figure 6.** Full time series water level data using multivariate input data.

**Table 7.** Comparison of multivariate model performance.

| Variables | Computational State | MSE (EL.m) | RMSE (EL.m) | $R^2$ | NSE |
|---|---|---|---|---|---|
| DWL, DFR, S_DVP, S_DDPT, S_1HP | Training | 0.0019 | 0.0437 | 0.9888 | 0.9923 |
| | Prediction | 0.0329 | 0.1814 | 0.4660 | 0.5976 |
| DWL, DFR, Y_DVP, Y_DDPT, Y_1HP | Training | 0.0022 | 0.0470 | 0.9843 | 0.9807 |
| | Prediction | 0.0326 | 0.1806 | 0.4896 | 0.5941 |
| DWL, DFR, * equal(S_DVP, S_DDPT, S_1HP, Y_DVP, Y_DDPT, Y_1HP) | Training | 0.0025 | 0.0497 | 0.9808 | 0.9860 |
| | Prediction | 0.0339 | 0.1840 | 0.4676 | 0.6107 |
| DWL, DFR, ** distance(S_DVP, S_DDPT, S_1HP, Y_DVP, Y_DDPT, Y_1HP) | Training | 0.0018 | 0.0420 | 0.9868 | 0.9868 |
| | Prediction | 0.0301 | 0.1734 | 0.5049 | 0.6228 |

* considering the data of Seoul Observatory and Yangpyeong Observatory equally; ** considering the weighted average of the data of the Seoul and Yangpyeong Observatories on the distance.

As shown in Table 3, the correlation coefficient between the meteorological data of each meteorological observatory and the DWL data was in the range of 0.2350 to 0.3254, which is not significant because the distance between the Paldang Bridge and each meteorological observatory is quite far away and geographically different from each meteorological observatory. As shown in Figure 6b, when DWL was predicted at Paldang Bridge using meteorological data from the Seoul Meteorological Observatory, some predicted DWL tended to be under-predicted compared to the measured DWL, and also under-estimated in the high water level section. In the case of Figure 6d, when predicting DWL using meteorological data from Yangpyeong Meteorological Observatory, the predicted DWL was slightly over-estimated in the low water level section, but relatively good prediction results were shown in the high water level section. It was evaluated that the Yangpyeong Meteorological Observatory (predicted $R^2 = 0.4896$) is located adjacent to the South Han River, thereby reflecting meteorological factors due to the influence of the river and in the case of Seoul Meteorological Observatory (predicted $R^2 = 0.4660$), it was installed at a long distance from the Han River and reflected meteorological factors in Seoul downtown.

However, for the eight multivariate data scenario which considered the meteorological data of Seoul Meteorological Observatory and Yangpyeong Meteorological Observatory at the same time, the predicted accuracy of DWL at Paldang Bridge station was improved to $NSE = 0.6107 - 0.6228$. According to Figure 6f, the prediction accuracy of the DWL predicted at the Paldang Bridge station was improved even if the meteorological data collected from the two meteorological observatories were considered with the same weight. In addition, as shown in Figure 6h, the prediction results obtained by weighted average of the meteorological data for the two meteorological observatories based on distance between the Paldang Bridge station and each meteorological observatory increased the prediction accuracy of the model to $NSE = 0.6228$. For Paldang Bridge station, the prediction point of DWL is located in the middle of the two meteorological observatories, the best DWL prediction result could be obtained if the observed meteorological data were trained by weighted average according to the distance between the prediction point and each meteorological observatory.

## 5. Discussion

In the case of other previous data-driven models [1–5,8–14], the prediction accuracy for rapidly changing peak water level fluctuations degraded. Various DNN models (CNN, simple RNN, LSTM, GRU) were also used to evaluate the accuracy of the models on the training and prediction results of low and high flow rates [15]. The LSTM and GRU model outputs yielded better prediction accuracy than other DNN models such as CNN and simple RNN at peak water levels. In the case of the GRU model, although its structure is

simpler than that of LSTM and computational cost was less, it had a slightly higher accuracy than that of LSTM. Therefore, this study used GRU to derive the results of this study.

Based on the results of the previous study [15], a GRU model for water level prediction was constructed using time series water level data. SL, a training unit of data necessary for learning hydrological data presented in the previous study, was used every 14 days. The ratio of the learning data and the prediction data of the hydrological data were divided into 74.9% and 25.1%, respectively, and constructed as input and output data of the GRU model.

Based on previous findings [21], multivariate DL models using five variables (DWL, DFR, DVP, DDPT, 1HP) were evaluated higher than univariate water level predictions. In this study, multivariate water level prediction was also performed using the distance-weighted average method on meteorological data from two observatories (i.e., Seoul and Yangpyeong Observatories) provided by the KMA. The univariate water level prediction result was $NSE = 0.5129$, but the accuracy was improved to $NSE = 0.6228$ in the multivariate model where a total of eight variables (i.e., DWL, DFR, S_DVP, S_DDPT, S_1HP, S_DDPT, Y_DDPT, Y_1HP) were learned as input data. In other words, if the distance from each meteorological observatory to the water level prediction point is far, but the distance-weighted average of the meteorological data of the two meteorological observatories is used, the accuracy of the water level prediction results will improve compared to that of the univariate water level results.

## 6. Conclusions

In the case of rivers flowing through the city coupled with the effect of climate change, accurate prediction of future flood water levels is very important. In this study, the GRU model which is excellent in multivariate prediction results was selected as a data-driven model suitable for predicting the water level of rivers with very large fluctuations in flood event. Using the GRU model, the time series water levels with large fluctuations were predicted.

In this study, DFR, hydrological data correlated with the DWL to be predicted, and DVP, DDPT, and 1HP, which are meteorological data of the meteorological observatory, were selected as learning data for the GRU model. When only univariate water level data were learned with the GRU model, the DWL predicted with the GRU model tended to be over-predicted compared to the actual observation data. Therefore, meteorological data (i.e., DVP, DDPT, and 1HP) of the meteorological observatory correlated with DWL and DFR were used as learning data to predict DWL with improved accuracy. However, when only meteorological data from a single meteorological observatory located at a distance from the water level prediction point are used, the predicted DWL was partially under-predicted or partially over-predicted, so the prediction accuracy of DWL did not improve.

This study proposed a prediction method of DWL with improved prediction accuracy even if the distance between the prediction point of DWL and the meteorological observatories is far and the meteorological data of each of the two meteorological observatories are not correlated. First, if the method of arithmetic average weather data of two or more meteorological observatories that can affect the prediction point, the prediction accuracy of the water level could be improved compared to the case of using the univariate water level or using the meteorological data of a single meteorological observatory. Second, the accuracy of the predicted DWL by multivariate learning would improve if the meteorological data of each meteorological observatory were used as training data of the GRU model by weighted average of the distance from the prediction point to the meteorological observatory.

The GRU model selected in this study is a data-driven model that analyzes the numerical variability of data, and it is possible to predict water levels with improved accuracy by using meteorological data considering the distance between the two meteorological observatories as multivariate learning data. Therefore, if meteorological data (i.e., DVP, DDPT, and 1HP) of two or more meteorological observatories located near the prediction point are used when predicting the rapidly changing water level, it is possible to predict the water level more accurately than the univariate water level prediction method.

**Author Contributions:** Conceptualization, K.P., Y.S. and Y.J.; methodology, K.P.; software, K.P.; validation, K.P.; formal analysis, K.P.; investigation, K.P., Y.S., Y.J., I.Y. and C.K.C.; resources, K.P., Y.S., Y.J., I.Y. and C.K.C.; data curation, K.P., Y.S., Y.J., I.Y. and C.K.C.; writing—original draft preparation, K.P.; writing—review and editing, K.P., Y.S., Y.J., I.Y. and C.K.C.; visualization, K.P.; supervision, Y.J. and I.Y.; project administration, Y.J.; funding acquisition, Y.J. All authors have read and agreed to the published version of the manuscript.

**Funding:** This research was supported by a grant (2022-MOIS61-001) of Development Risk Prediction Technology of Storm and Flood for Climate Change based on Artificial Intelligence funded by Ministry of Interior and Safety (MOIS, Korea).

**Institutional Review Board Statement:** Not applicable.

**Informed Consent Statement:** Not applicable.

**Data Availability Statement:** Data for this work can be found within the article; for further data, please contact the first and corresponding authors.

**Acknowledgments:** Park, K., Jung, Y., Seong, Y., and Youn, I. acknowledge the financial support of the Emergency Management Institute at Kyungpook National University, the Department of Advanced Science and Technology Convergence at Kyungpook National University, and Department of Construction and Disaster Prevention Engineering, Kyungpook National University.

**Conflicts of Interest:** The authors declare no conflict of interest.

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
