# Peer review of "Development of Water Level Prediction Improvement Method Using Multivariate Time Series Data by GRU Model"

_water, doi:10.3390/w15030587_

Round 1

Reviewer 1 Report

Based on the GRU method, this study predicted the water level at the Paldang Bridge station. Frankly, the study has so many problems overall.

Basic issues: There are lots of detail flaws, such as Figure 3 in 185 line should be Figure 2, and variables annotations should be in order in table 3.

Logic problems: why cant we just interpolate or utilize the raster data as input variables instead of using the distance-weighted station data?

Methodological problems:

(1) the results of the model fitting exhibit obvious over-fitted feathers. Second, what does the better method mean when the accuracy improvement of R²is just 0.04 and the NSE is 0.1?

(2) Since there is such a weak correlation between meteorological data and water level, why are they still necessary to be included in the machine learning model? 

Author Response

We highly appreciate the valuable comments given by you on our work. We addressed each of them in this rebuttal as well as in the revised manuscript.

Reviewer 2 Report

The manuscript tackles a complex and interesting topic but as presented in my opinion does not provide any scientific output. The manuscript has to be thoroughly revised and improved.

The introduction is poor and with a number of questionable assertions.

Some sentences are complex. For example: "In the past, Computational Fluid Dynamics (CFD) based physical models based on dynamic dynamics were widely used in terms of river planning and disaster prevention rather than using data driven 51 models." Not only for the grammar but also for its meaning. I don´t know many models using CFD for river planning or disaster prevention.

I don´t agree with those sentences: "In order for the data driven model to be used for hydrological analysis and prediction, it is required to improve the calculation speed and prediction accuracy that can replace the existing dynamic physics model."

I suggest the authors to deeply modify the introduction to 1) review the existing studies, 2) point out the gaps to be filled and 3) define the way the contribute to fill the gaps with the works they are presenting.

The method is clear enough for me but I think the authors should clearly explain the reason why this method is suitable for solving the problem they are addressing. FI don’t understand why the authors present first the water level data and second Hydrological and meteorological data. Does it mean water level data are not hydrological data?

The results section is not clear for me and it seems the authors merely present the results from running the algorithm in both the training and validation datasets. In my opinioj it does not provide relevant outputs.

Author Response

(The authors gave the same response as above.)

Reviewer 3 Report

The article presented something interesting on river engineering management for water operation and sustainability. Here are my concerns:

> The research gap not clearly presented, why this model selected why this case study investigated?

> Is it suffcient to test one single dataset?

> What about the modeling validation with benchmark models.?

> How did you validated the future forecast?

> The methodology is very slightly presented more details are needed.

> More graphical presentation for the presented results must be explained.

> Models likes SVM, ANFIS, ANN are needed to verify the modeling performance.

> More practical discussion is needed to satisfy the concept of water engineering.

Author Response

(The authors gave the same response as above.)

Round 2

Reviewer 2 Report

The authors have addressed my previous comments.

Author Response

We highly appreciate the valuable comments given by you on our work. We addressed each of them in this rebuttal as well as in the revised manuscript.

As mentioned by two reviewers, the English correction through native speakers was completed. The modified words or sentences are marked in red.

Reviewer 3 Report

The authors have revised the manuscript in good scientific manner.

Author Response

(The authors gave the same response as above.)
